

# Sports experts' unique perception of time duration based on the processing principle of an integrated model of timing

Binbin Jia[1], Zhongqiu Zhang[2] and Tian Feng[3]

[1] School of Psychology, Shanghai University of Sport, Shanghai, China
[2] Sports Psychology and Biomechanics Research Center, China Institute of Sport Science, Beijing, China
[3] Physical Education College of Zhengzhou University, Zhengzhou, China

## ABSTRACT

**Background**. Duration perception is an essential part of our cognitive and behavioral system, helping us interact with the outside world. An integrated model of timing, which states that the perceived duration of a given stimulus is based on the efficiency of information extraction, was recently set forth to improve current understanding of the representation and judgment of time. However, the prediction from this model that more efficient information extraction results in longer perceived duration has not been tested. Thus, the aim of this study is to investigate whether sports experts, as a group of individuals with information extraction superiority in situations relevant to their sport skill, have longer duration perceptions when they view expertise-related stimuli compared with others with no expertise/experience.

**Methods**. For this study, 81 subjects were recruited based on a prior power analysis. The sports experts group had 27 athletes with years of professional training in diving; a wrestler group and a nonathlete group, with each of these groups having 27 subjects, were used as controls. All participants completed a classic duration reproduction task for subsecond and suprasecond durations with both the diving images and general images involved.

**Results**. The divers reproduced longer durations for diving stimuli compared with general stimuli under both subsecond and suprasecond time ranges, while the other samples showed the opposite pattern. Furthermore, the years of training in diving were positively correlated with the magnitude of the prolonged reproduction duration when divers viewed diving stimuli. Moreover, the diver group showed a more precise duration perception in subsecond time range for general stimuli compared with the wrestlers and nonathletes.

**Conclusion**. The results suggest that sports experts perceive longer duration when viewing expertise-related stimuli compared with others with no expertise/experience.

Corresponding author
Binbin Jia, jbb1126@outlook.com

## INTRODUCTION

The experience of time is fundamental for how we make sense of the world. Duration perception, a component of time perception, is a "basic unit of ability" on which other cognitive and behavioral functions are based (*Allman & Meck, 2012*; *Buhusi & Meck, 2005*; *Mauk & Buonomano, 2004*). Due to the importance of duration perception, it has been intensively studied for many centuries (*Hancock & Block, 2012*), and various models have emerged to explore the underlying mechanism (*Buonomano, Bramen & Khodadadifar, 2009*; *Buonomano & Maass, 2009*; *Eagleman & Pariyadath, 2009*; *Matell & Meck, 2004*; *Noguchi & Kakigi, 2006*; *Treisman, 1963*; *Walsh, 2003*). However, no single model is sufficient to explain all facets of duration perception (*Ivry & Schlerf, 2008*; *Matthews & Meck, 2016*). Therefore, an integrated framework combining the strength of different models will undoubtedly be welcome.

*Matthews & Meck (2016)* integrated current findings and models in the duration perception domain to provide an integrated model called the processing principle. The authors suggested that perceived duration is positively related to perceptual vividity and the ease of extracting information from stimuli. In other words, the greater the amount of information is extracted from a given stimulus, the longer the perceived duration. Based on this proposal, sports experts, as a group with efficient information extraction in expertise-related tasks owing to their well-researched cognitive advantage (*Ericsson & Kintsch, 1995*; *Ericsson & Lehmann, 1996*; *Feng et al., 2017*; *He et al., 2018*; *Wei & Luo, 2010*; *Yarrow, Brown & Krakauer, 2009*), should perceive a longer duration than people who lack such expertise when they view an expertise-related stimulus. However, this prediction has not yet been tested. Moreover, although some studies have suggested that sports experts have more precise and stable duration perception than nonathletes (*Chen & Cesari, 2015*; *Chen, Pizzolato & Cesari, 2013*; *Chen, Pizzolato & Cesari, 2014*), whether sports experts perceive a longer duration than others for expertise-related stimulus remains unclear.

The present study aims to investigate sports experts' duration perception for both expertise-related stimuli and general stimuli. Thus, we recruited divers, wrestlers and nonathletes to complete a classic duration reproduction task to evaluate the participants' duration perception of expertise-related stimuli (diving movements). General stimuli (geometric figures) were presented during the task to obtain a baseline measure. We chose divers as participants for two reasons. First, divers in China compete at a high level (*Wei & Luo, 2010*), which makes them suitable to serve as representative sports experts. Second, duration perception is important for divers, since they need to perform complex movements in situations with high time pressure. Wrestlers with no diving experience but long-term physical training were recruited as a control group, and nonathletes with no diving experience or long-term physical training experience were also recruited as a control group. Moreover, duration perception in both subsecond and suprasecond time ranges was examined during this study. Previous studies have suggested that suprasecond time range is affected more than subsecond range by cognition (*Hayashi et al., 2014*; *Lewis & Miall, 2003a*; *Lewis & Miall, 2003b*; *Rammsayer & Ulrich, 2011*; *Rammsayer & Troche, 2014*). Therefore, we assume that sports experts' information extraction advantage will
be expressed more in suprasecond time range than in subsecond time range, which is mainly ruled by automatic processes. Our specific hypothesis is that divers, compared with wrestlers and nonathletes, will show lengthened duration perception for diving-related stimuli in suprasecond time range.

## MATERIAL AND METHODS

This study received approval from the regional ethics board of the China Institute of Sport Science in China (approval number: 18-04). Written informed consent was obtained from all participants before the experiment. Importantly, all the materials involved in this research can be found at osf.io/83smd/ (i.e., the task procedure, stimuli, participants' demographic information, raw data, and statistical results).

### Participants

The sample size of this study was set at 81 according to a priori power analysis using G*Power software (*Franz et al., 2007*). Specifically, four effects (2 main effects and 2 interaction effects) based on our hypothesis and previous studies were examined in this study. The power calculations for these effects were based on the F-tests (repeated-measures ANOVA); the α value was set at 0.05, the statistical power (1-β) was set at 0.8, the correlation among repeated measures was set at 0.5, and the nonsphericity correction was set at 1. The effect size f for the effects that had been explored was determined according to previous studies (*Chen & Cesari, 2015*; *Chen, Pizzolato & Cesari, 2013*), and a small effect size was assumed based on Cohen's approaches for unexplored effects (*Cohen, 1988*). Moreover, due to considerations of publication bias and a tendency to overestimate effect size in underpowered studies (*Schafer & Schwarz, 2019*; *Schweizer & Furley, 2016*), the effect size in our power analysis was half the magnitude of the original effects based on the findings from Open Science Collaboration (*Open Science, 2015*). Ultimately, the largest sample size that came from these calculations was chosen to conduct the experiment. The sample size determination process is shown in Fig. 1.

The sports experts group had 27 divers (15 female, years of training 8.06 ± 2.84, age 14 ± 3.09) based on the criteria of being at least National Level 2 athletes (ranking among top 3 in provincial competitions) and actively participating in national-level diving competitions. The amateur athletes group contained 27 wrestlers (3 female, years of training 2.94 ± 1.63, age 16.52 ± 1.22) from a sports academy, all of them were active in provincial-level competitions. The nonathlete group consisted of 27 junior school students (13 female, age 12.56 ± 0.51), none of them had any experience in sports at competitive level. All participants had normal or corrected-to-normal vision and had no knowledge of the purpose of the experiment.

### Task

In the current study, a classic duration reproduction task was utilized (*Chen & Cesari, 2015*). During the task, the participants were first presented with a fixation target; then, a visual stimulus was shown for a specific duration. The participants were told to remember the duration of a stimulus, and after a brief delay, they need to press and hold the space bar of a computer keyboard for the same duration using their index finger (Fig. 2).
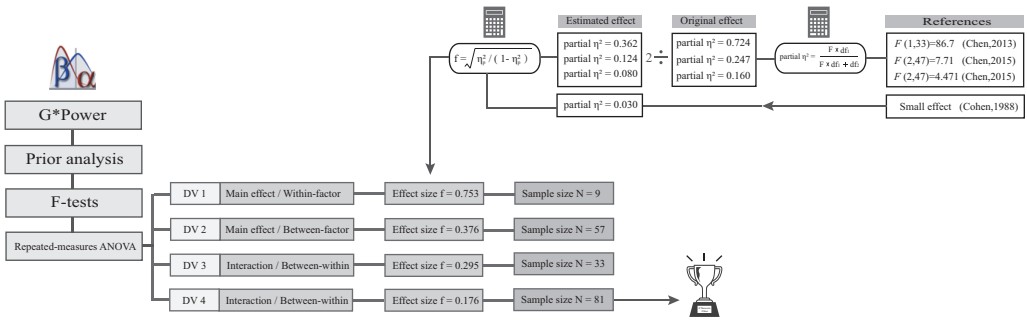

**Figure 1 The sample size determination process.** DV1, dependent variable 1; DV2, dependent variable 2; DV3, dependent variable 3; DV4, dependent variable 4. The $F$ values from previous studies were provided to calculate the partial eta square (partial $\eta^2$). Then, half the magnitude of the partial $\eta^2$ was translated into effect size f, which is needed to calculate the sample size in G*Power. Partial $\eta^2 = 0.030$ was chosen to determine the sample size for unexplored effect (DV4).

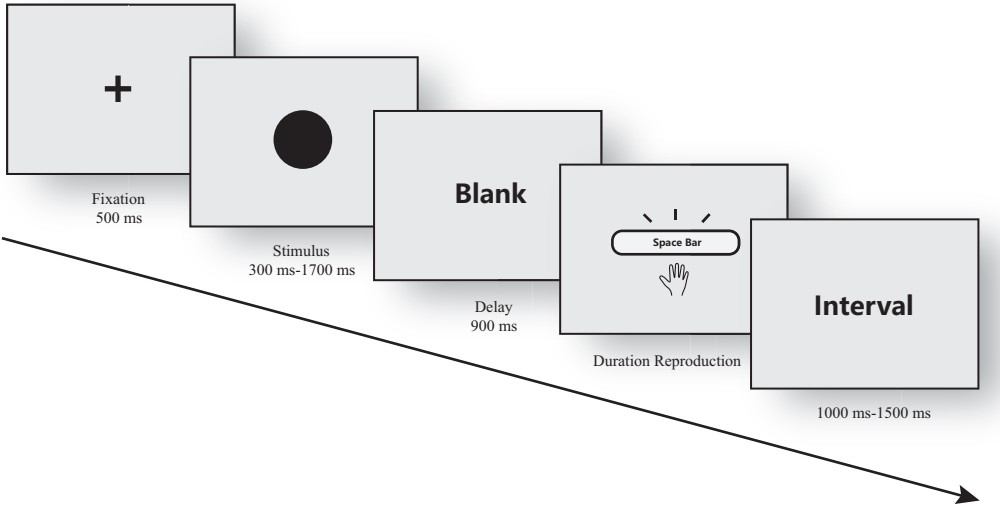

**Figure 2 Temporal duration reproduction task.** A sequence within a single trial of the temporal duration reproduction task in this experiment.

## Materials and procedures

Two types of stimuli were employed in this task: general stimuli and expertise-related stimuli. The general stimuli, including a square, a circle, a star, and a triangle, were created in Adobe Illustrator CS6 (Adobe Systems Incorporated, San Jose, CA, USA); each one was 27 mm tall, and all of them were black in color (RGB: 0, 0, 0). The expertise-related stimuli (four diving movements), created with the help of four Chinese national divers, were selected from videos recorded with a Sony camera (HXR-NX3) during the divers' regular practice at the national diving pool facility. The initial images were then re-edited using Adobe Illustrator CS6 to remove the background. Different stimuli were used to control the anticipation and adaptation effects for duration perception (*Matthews & Meck, 2016*). In addition, six different durations were used for the stimulus presentations: 300

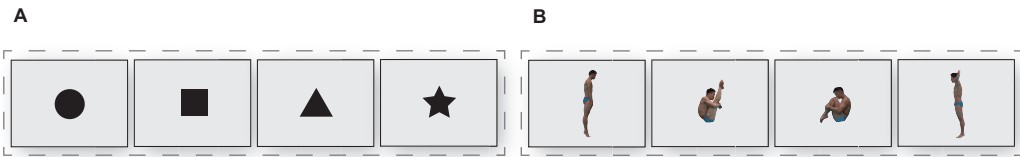

**Figure 3 All stimuli in the task.** (A) Stimuli in the general condition; (B) stimuli in the expertise-related stimuli condition.

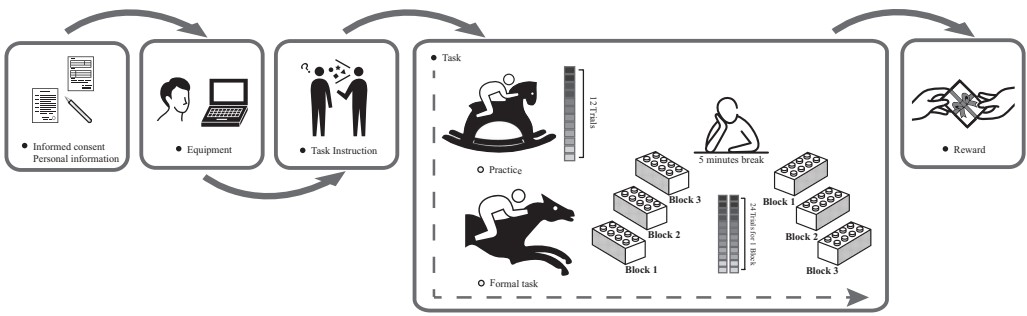

**Figure 4 The experimental process.**

ms, 500 ms, and 700 ms for subsecond time range and 1,300 ms, 1,500 ms and 1,700 ms for suprasecond time range. All stimuli used in the task are shown in Fig. 3.

The task was designed and run using E-Prime 1.1 (Psychology Software Tools, Pittsburgh, PA, USA). A laptop with a 15.6-inch screen (1024 × 768, 59 Hz) was used to run the experimental procedure. The experiment was conducted in a room isolated from external lights and noises. The participants were seated in front of the laptop, approximately 70 cm away from the screen, and were given instructions for the task. The subjects were also told that counting strategies were not allowed during the task (*Rattat & Droit-Volet, 2012*). Then, the subjects performed a short practice session with feedback (12 trials) to ensure that they fully understood the task. After practice, the subjects were prompted to initialize the formal experiment, in which no feedback was given. The formal task was separated into 2 sessions, one for the general stimuli and another for the expertise-related stimuli. Each session had 3 blocks interspersed with 2 short breaks. Each block had 24 trials (4 stimuli × 6 durations). The order of the combinations of stimuli and durations was pseudorandomized during each block. The participants were given a 5-minute break when the first session was completed. The order of the sessions was counterbalanced within the subjects. All participants were rewarded with gifts when the experiment was completed. The entire experimental process is presented in Fig. 4.

## Data analysis and statistics

The raw data were fed into R (http://www.r-project.org) for preprocessing and plotting (*R Core Team, 2018*). Six participants (three from the wrestler group, three from the nonathlete group) were removed from further analysis because they misunderstood the task or were

unwilling to cooperate (more than 50% of reproduction durations ≤300 ms, 73.6% overall), and six participants were added as replacements to achieve the predetermined sample size. Individuals with reaction times greater than 2,000 ms during the reproduction phase were excluded (3.4% of the data were rejected). Then, medians for every participant under different conditions were calculated for subsequent statistical analysis. The raw data for different groups were converted to raincloud plots to obtain robust data visualization (*Allen et al., 2018*).

The dependent variables were as follows: (1) The reproduction duration (RD) was used to investigate the participants' duration perception under different conditions during the task.

(2) The reproduction duration difference (RDD), defined as the RD for expertise-related stimuli minus the RD for general stimuli (baseline measure), was used to remove interindividual differences during the task. (3) Estimation bias (hereafter referred to as bias) was calculated based on the underestimation percentage (the percentage of trials for which stimulus duration $-$RD $< 0$) in the task. (4) The coefficient of variation (CV) of the RD, based on the ratio of the standard deviation to the mean, was used to investigate the participants' duration perception stability. (5) The absolute error in the ratio (AE ratio) to the corresponding stimulus duration was used to evaluate the precision of participants' duration perception. The RD, RDD and bias help us answer the question of whether divers perceive longer durations than wrestlers or nonathletes for stimuli depicting diving movements. If so, the RDD of the divers will be larger than that of the wrestlers and nonathletes in the suprasecond time range, and the bias of divers will be smaller (less underestimated) when they view diving movement stimuli than general stimuli, while no such difference will appear in wrestlers or nonathletes. The CV and AE ratio were targeted here to replicate the findings from previous studies, which suggest that sports experts have more stable and accurate duration perception (a smaller CV and AE ratio) than nonathletes.

The statistical analyses were conducted using JASP 0.10.0.0 (http://www.jasp-stats.org). A two-way repeated-measures ANOVA (with group as the between-subjects factor, time range as the within-subjects factor, and age as the covariate) was used for the RDD. In addition, a three-way repeated-measures ANOVA (with group as the between-subjects factor, time range and image type as within-subjects factors, and age as the covariate) was employed for the RD, bias, CV, and AE ratio. For each of the statistical tests involved, 95% confidence interval (CI) for effect size (partial $\eta^2$ in $F$-test and Cohen's d in $t$-test), Bayesian factor ($BF_{10}$) and $p$-value are provided. The Bayesian factors were calculated via Bayesian statistics functions in JASP to quantify the strength of evidence for both the null hypothesis and alternative hypothesis (*Wagenmakers et al., 2018*). The prior of the Bayes factor ANOVA is a default distribution in JASP and combines the multivariate and independent Cauchy distributions (*Rouder et al., 2012*), and the model parameters for Bayes factor ANOVA are r scale for fixed effects $= 0.5$, random effects $= 1$, r covariates $= 0.354$. Additionally, a uniform distribution prior for Bayesian Correlation Matrix with stretched beta prior width $= 1$. Finally, to enable other researchers to replicate the statistical results, we provided the full statistical results of the Bayes factor ANOVA at osf.io/83smd.

More importantly, we also ran statistical analyses considering the difference in years of training between the divers and wrestlers ($8.06 \pm 2.84$ versus $2.94 \pm 1.63$) and gender difference between the wrestlers (only 3 females) and the other groups. The main aim was to evaluate whether our results were influenced by those factors. Therefore, we analyzed the relationship of years of training with RD and RDD in both divers and wrestlers. Moreover, we selected subsets of divers ($n = 10$) and wrestlers ($n = 10$) with comparable durations of training ($5.3 \pm 1.57$ versus $4.55 \pm 1.52$, $t(18) = 1.087$, $p = 0.291$, Cohen's $d = 0.486$) to examine the group difference with regard to the RDD results. Additionally, to evaluate the effect of gender difference on our results, we ran repeated-measures ANOVAs of RD, RDD, bias, CV, and AE ratio results for divers ($n = 27$, 15 female) and non-athletes ($n = 27$, 13 female) with both group and gender as the between-subjects factors, time range as the within-subjects factor, and age as the covariate.

# RESULTS

## RD

The three-way repeated-measures ANOVA detected an interaction between image type and group, $F(2, 77) = 4.066$, $p = 0.021$, partial $\eta^2 = 0.096$ (90% CI [0.008–0.192]), and $BF_{10} = 2.63$. The divers reproduced longer durations for expertise-related stimuli compared with general stimuli in both subsecond and suprasecond time ranges, while the wrestlers and nonathletes showed the opposite result (Figs. 5D–5F). The group-time range-image type interaction did not reach the significance level ($\alpha = 0.05$), $F(2, 77) = 0.556$, $p = 0.576$, partial $\eta^2 = 0.014$ (90% CI [0.000–0.066]), and $BF_{10} = 0.132$ ($BF_{01} = 7.576$).

## RDD

The two-way repeated-measures ANOVA demonstrated a main effect of group in terms of the RDD, $F(2, 77) = 4.066$, $p = 0.021$, partial $\eta^2 = 0.096$ (90% CI [0.008–0.192]), and $BF_{10} = 1.428$. The results showed that compared with the other two groups, the divers had large RDD values in both subsecond and suprasecond time ranges (Fig. 6E). A comparison of three groups revealed the following results with regard to the divers and wrestlers: $t = 2.830$, Cohen's $d = 0.314$ (95% CI [0.213–1.320]), $p_{holm} = 0.018$, and $BF_{10} = 11.668$. The results of the comparison between the divers and nonathlete controls were as follows: $t = 1.132$, Cohen's $d = 0.126$ (95% CI [−0.230–0.843]), $p_{holm} = 0.261$, and $BF_{10} = 1.646$.

## RDD and years of diving training

To explore the relationship between the RDD and years of diving training among the divers, we used the Pearson correlation coefficient. The results showed that in subsecond time range, $r = 0.315$ (95% CI [−0.083–0.626]), $p = 0.117$, and $BF_{10} = 0.780$ ($BF_{01} = 1.282$); additionally, in suprasecond time range, $r = 0.589$ (95% CI [0.262–0.795]), $p = 0.002$, and $BF_{10} = 28.127$. The longer the divers' training experience in diving, the larger their RDD, especially in suprasecond time range. The raw data are shown in Fig. 7. One diver's data were excluded from statistical analysis because of the extreme value (<mean −3.5*SD in the suprasecond condition); the results including this extreme value are $r = 0.198$ (95% CI [−0.197–0.538]), $p = 0.322$, and $BF_{10} = 0.381$ ($BF_{01} = 2.625$) in subsecond time range

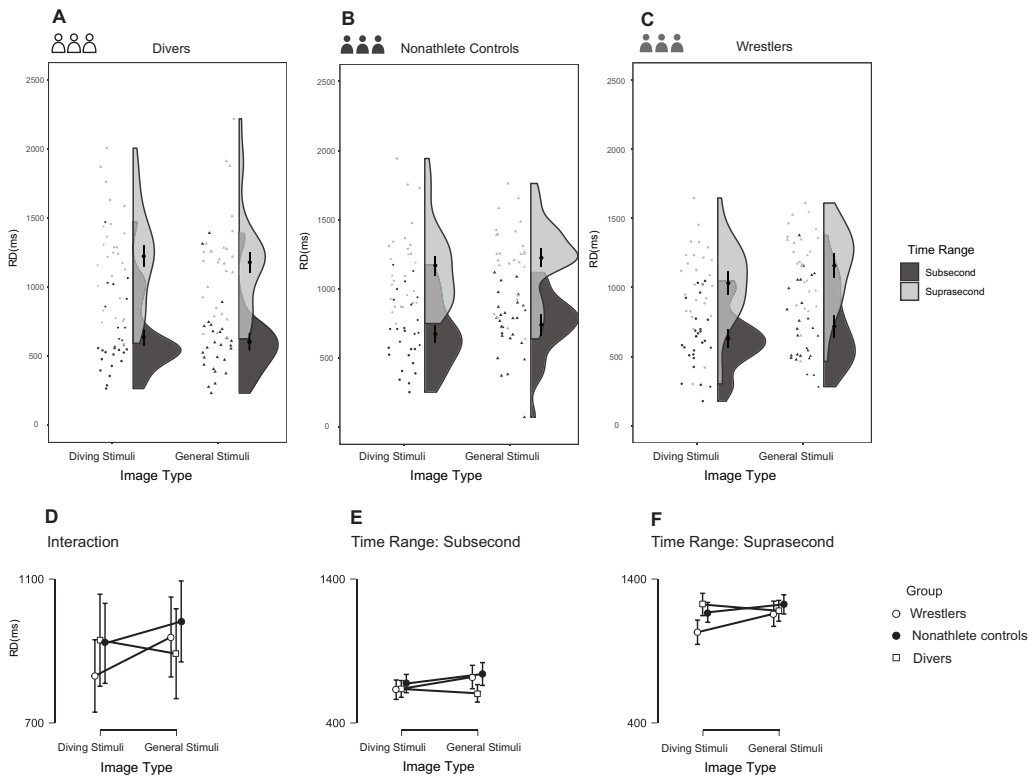

**Figure 5** **Reproduction duration in the three groups during the task.** (A–C) The raw data with distribution for different groups are provided via raincloud plots. The error bar indicates 95%CI of the mean; (D–F) Comparisons between different groups under different conditions (i.e., subsecond time range and suprasecond time range) are also presented.

and $r = 0.283$ (95% CI [$-0.108$–$0.599$]), $p = 0.152$, and $BF_{10} = 0.633$ ($BF_{01} = 1.58$) in suprasecond time range.

## Bias

As shown in Figs. 8A–8C, the three-way repeated-measures ANOVA detected a main effect of time range, $F(1, 77) = 8.866$, $p = 0.004$, partial $\eta^2 = 0.103$ (90% CI [0.020–0.218]), and $BF_{10} = 8.954e+59$, with most of the participants in all three groups overestimating (bias < 0.5) more at subsecond durations and underestimating (bias > 0.5) more at suprasecond durations. Moreover, an interaction effect between image type and group was found, $F(2, 77) = 5.338$, $p = 0.007$, partial $\eta^2 = 0.122$ (90% CI [0.021–0.228]), and $BF_{10} = 1.963$ (Fig. 8D). This interaction effect revealed that the divers overestimated more in the diving image condition than in the general image condition, while the other participants showed the opposite result.

## CV

As illustrated in Fig. 9D, the three-way repeated-measures ANOVA detected an interaction of time range and age (covariate), $F(1, 76) = 6.975$, $p = 0.010$, partial $\eta^2 = 0.084$ (90% CI [0.012–0.195]), and $BF_{10} = 3.907e+20$. The results showed that the participants in all

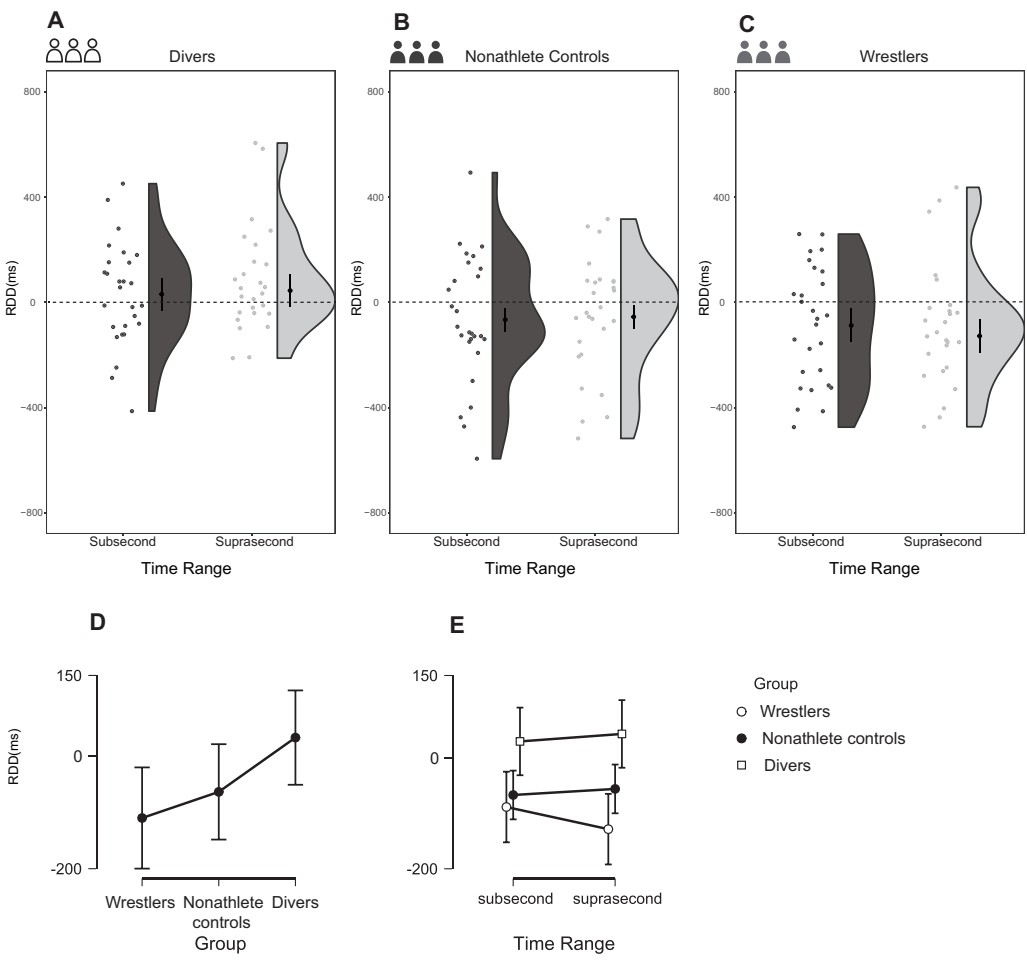

**Figure 6** **Reproduction duration differences in the three groups during the task.** (A–C) The raw data with distribution for different groups; (D–E) Comparisons between different groups under different conditions.

three groups had a more stable reproduction duration in suprasecond time range than in subsecond time range, $t = 11.597$, Cohen's $d = 1.297$ (95% CI [0.996–1.592]), $p < 0.0001$. In addition, the main effect of group (Figs. 9E–9F) did not reach the significance level ($\alpha = 0.05$), $F(2, 76) = 0.701$, $p = 0.499$, partial $\eta^2 = 0.018$ (90% CI [0.000–0.073]), and $BF_{10} = 0.117$ ($BF_{01} = 8.547$), nor did the interaction effect between group and time range, $F(2, 76) = 2.263$, $p = 0.111$, partial $\eta^2 = 0.056$ (90% CI [0.000–0.079]), and $BF_{10} = 0.073$ ($BF_{01} = 13.699$). One wrestler's CV was removed from the data plot and ANOVA due to the extreme value (>mean+8*SD in the diving image condition); the ANOVA results including this extreme value are $F(1, 77) = 4.751$, $p = 0.032$, partial $\eta^2 = 0.058$ (90% CI [0.002–0.159]), and $BF_{10} = 1.979e+6$ for the interaction of time range-age. The results for the main effect of group were $F(2, 77) = 0.619$, $p = 0.541$, partial $\eta^2 = 0.016$ (90% CI [0.000–0.068]), and $BF_{10} = 0.169$ ($BF_{01} = 5.917$), and those for the interaction effect

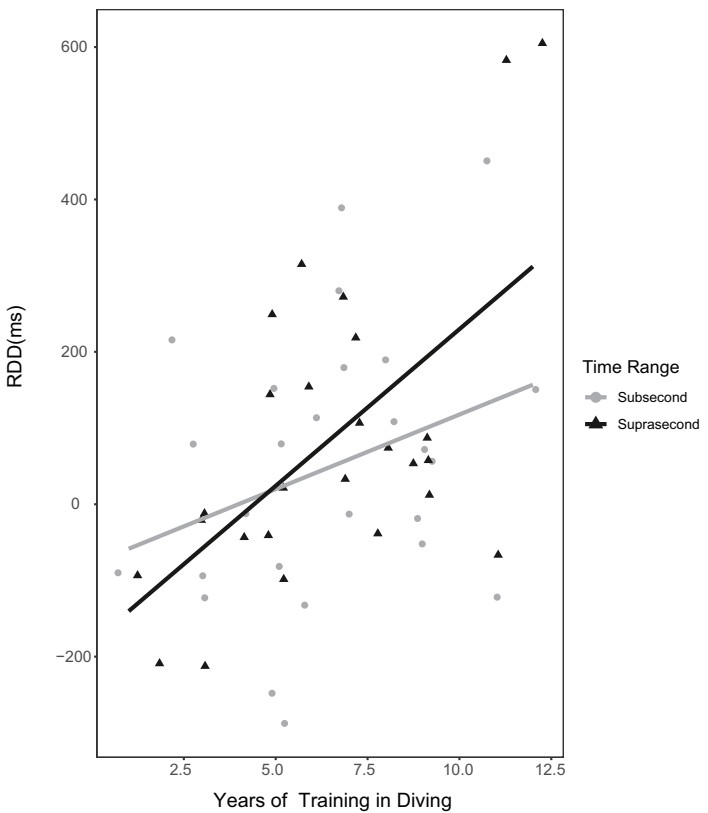

**Figure 7 The correlation between the RDD and years of diving training of the divers.** The lines in this plot indicate the correlation between variables based on the least squares method under different conditions (subsecond and suprasecond).

between group and time range were $F$ (2, 77) = 0.831, $p = 0.440$, partial $\eta^2 = 0.021$ (90% CI [0.000–0.082]), and $BF_{10} = 0.060$ ($BF_{01} = 16.667$).

## AE Ratio

The three-way repeated-measures ANOVA detected a main effect of time range (Fig. 10D), $F$ (1, 77) = 6.544, $p = 0.012$, partial $\eta^2 = 0.078$ (90% CI [0.010–0.187]), and $BF_{10} = 7.477e+11$, with most of the participants having a larger AE ratio in subsecond time range than in suprasecond time range. Moreover, an interaction effect of time range, image type, and group was revealed (Figs. 10E–10F), $F$(2, 77) = 5.254, $p = 0.007$, partial $\eta^2 = 0.120$ (90% CI [0.020–0.226]), and $BF_{10} = 0.874$ ($BF_{01} = 1.144$), revealing that the diver group produced smaller AE ratios than the other groups in subsecond time range under the general image condition.

## Effects of years of training and gender

To explore the relationship of years of training with RD and RDD in wrestlers and divers, we used the Pearson correlation coefficient. The results showed no evidence that years of training affected RD for diving images in wrestlers or divers (Figs. 11A–11B). Specifically, $r = -0.107$ (95% CI [−0.468–0.285]), $p = 0.596$ in subsecond time range and $r = -0.055$

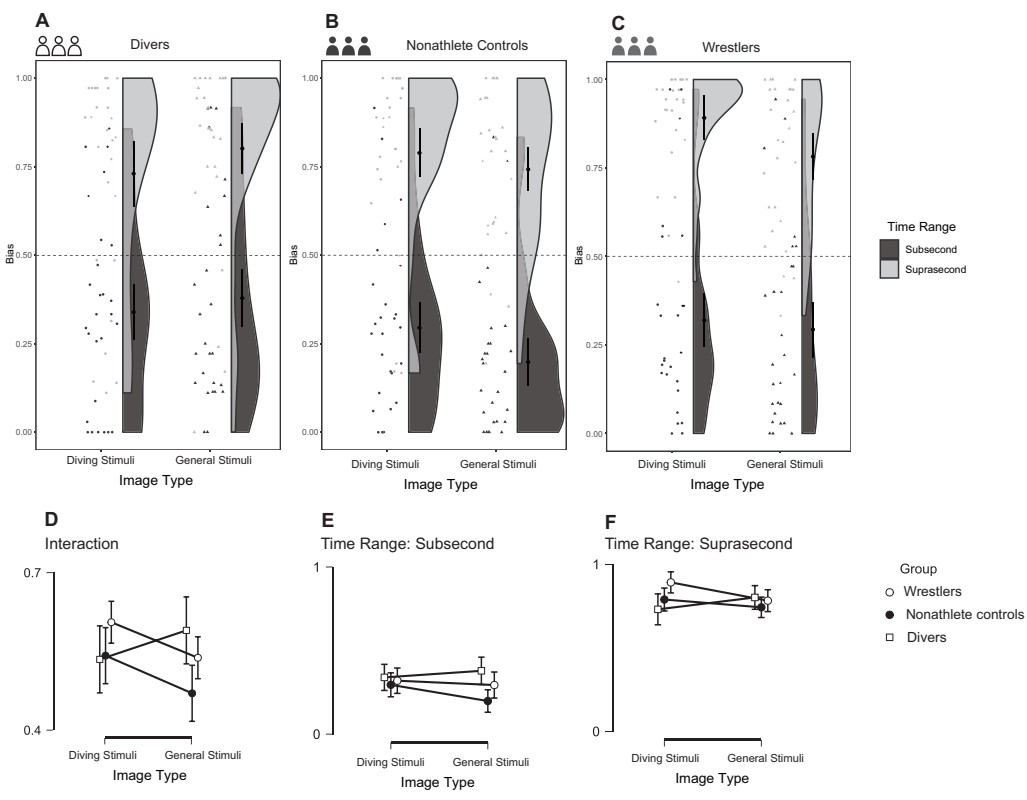

**Figure 8 Reproduction duration bias in the three groups during the task.** (A–C) The raw data with distribution for different groups; (D) the interaction of group and time range; (E–F) comparisons between different groups under different conditions.

(95% CI [−0.426–0.332]), $p = 0.785$ in suprasecond time range for wrestlers; $r = −0.207$ (95% CI [−0.55–0.196]), $p = 0.31$ in subsecond time range and $r = −0.059$ (95% CI [−0.437–0.336]), $p = 0.774$ in suprasecond time range for divers. Additionally, there was no evidence supporting a relationship between years of training and RDD in wrestlers (Fig. 11C). We found that $r = −0.011$, $p = 0.957$ (95% CI [−0.389–0.371]) in subsecond time range and $r = 0.153$ (95% CI [−0.241–0.503]), $p = 0.447$ in suprasecond time range for the wrestlers. More importantly, the relationship between years of training and RDD increased for divers when we considered only expertise-related training experience (Figs. 11D–11E). For example, one diver had 8 years of training experience, including 3 years of training for basketball and 5 years of training for diving. When we ran the analysis between years of diving training and RDD, r increased from 0.133 (all training) to 0.315 (only diving training) in subsecond time range and increased from 0.433 to 0.589 in suprasecond time range. Finally, the RDD group difference remained (divers > wrestlers) when we compared participants in both groups with compatible lengths of training (Figs. 11F–11G). Specifically, the statistical results were t(18) = 2.426, $p = 0.026$, Cohen's $d = 1.085$ (95% CI [0.127–2.016]) in subsecond time range and t(18) = 2.128, $p = 0.047$, Cohen's $d = 0.952$ (95% CI [0.011–1.869]) in suprasecond time range.

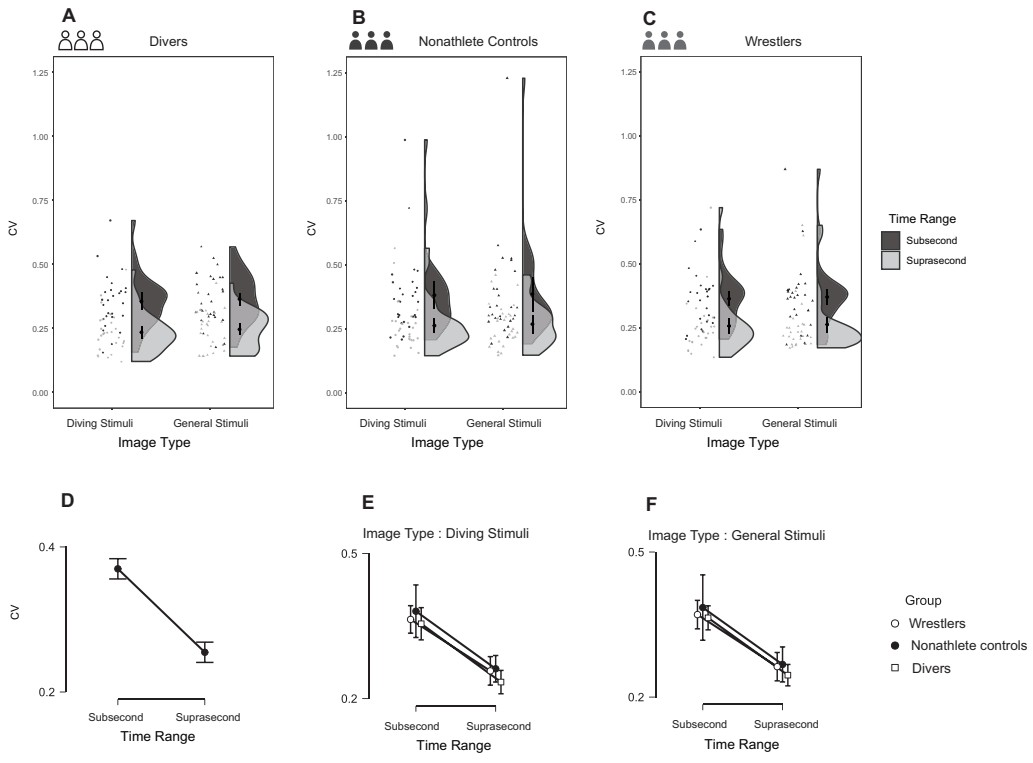

**Figure 9** **Reproduction duration CV in the three groups during the task.** (A–C) The raw data with distribution for different groups; (D) the comparison between different time range; (E–F) comparisons between different groups under different conditions.

Finally, ANOVA with an effect of gender revealed statistically interactions of time range and gender on bias ($F(1,48) = 6.381$, $p = 0.015$) and AE ratio ($F(1,48) = 4.37$, $p = 0.042$). However, no other results (whether main effects or interactions) reached the significance level ($\alpha = 0.05$) in RD, RDD and CV.

## DISCUSSION

The main purpose of this study was to investigate the duration perception of sports experts. Based on the processing principle, the initial prediction was that the divers would reproduce longer durations than wrestlers or nonathletes for expertise-related stimuli (diving movements) in suprasecond time range. The RD, RDD and bias results are consistent with this hypothesis.

According to the RD and RDD results, the divers reproduced longer durations when viewing diving movements than general stimuli in both subsecond and suprasecond ranges, while the wrestlers and nonathletes showed the opposite result. First, the divers' increased duration perception for expertise-related stimuli compared with general stimuli in suprasecond time range is in line with our prediction. We attribute this phenomenon to sports experts' efficiency advantage in extracting information from expertise-related stimuli. *Wei & Luo (2010)* found that divers utilized kinesthetic imagery more efficiently

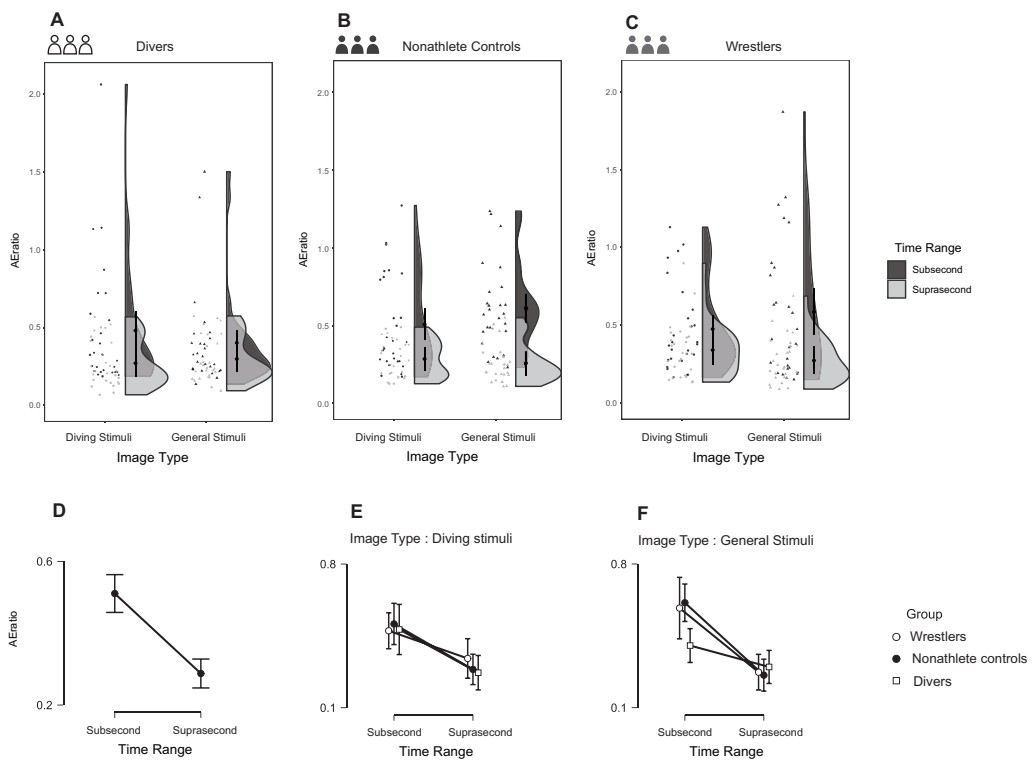

**Figure 10 Reproduction duration AE ratio in the three groups during the current task.** (A–C) The raw data with distribution for different groups; (D) the comparison between different time range; (E–F) comparisons between different groups under different conditions.

than novices for activities in which they had expertise. Sports experts' improved information extraction efficiency might stem from their cognitive advantages, which have been well demonstrated by many researchers. For instance, a review by *Yarrow, Brown & Krakauer (2009)* concluded that sports experts show superior performance in perception, anticipation and decision making and this superior performance is task specific and dependent on extensive practice. In addition, many other studies have revealed that sports experts outperform nonathletes in attention and memory for expertise-related tasks (*Ericsson & Kintsch, 1995*; *He et al., 2018*). Consequently, the advantages that sports experts hold in perception, attention and memory facilitate the extraction of information from expertise-related stimuli, causing them to perceive longer durations than nonathletes. In line with this suggestion, some studies have already found that as a result of better attention and memory, duration perception lasts longer (*Baudouin et al., 2006*; *Seifried & Ulrich, 2011*).

Meanwhile, many findings from the domain of time perception indicate that different temporal information processes and neural systems are involved in different time ranges (*Hayashi et al., 2014*; *Lewis & Miall, 2003a*; *Lewis & Miall, 2003b*; *Rammsayer & Ulrich, 2011*). Specifically, a cognitively controlled mechanism to process durations in suprasecond range and a sensory or automatic mechanism to process durations in subsecond range. For example, *Hayashi et al. (2014)* found that subsecond durations are processed in the

Peer J

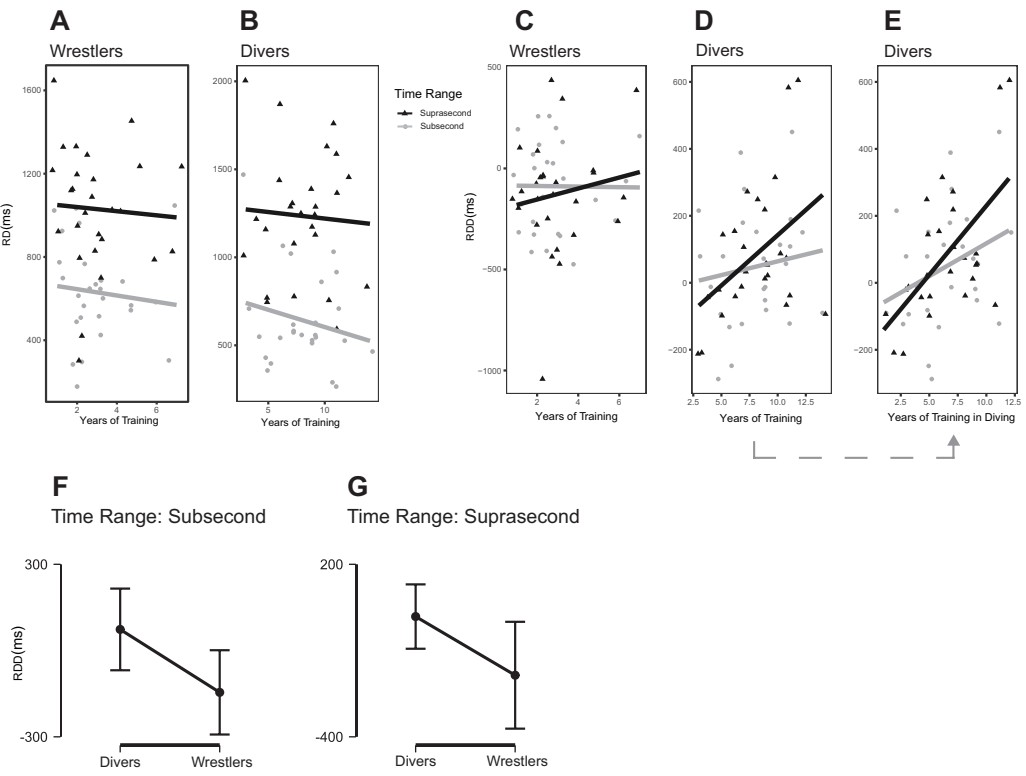

**Figure 11** **The analyses for effect of years of training difference between divers and wrestlers.** (A–B) The correlations between years of training and RD for diving figures in wrestlers and divers; (C–E) the correlations between years of training and RDD for wrestlers and divers; (F–G) comparison between divers and wrestlers with compatible years of training (error bar = mean ± 95% CI).

motor system, whereas suprasecond durations are processed in the parietal cortex using attention and working memory. Therefore, combined with athletes' cognitive advantages in expertise-related tasks and the larger role of cognition at longer durations, the increase in the perceived duration of suprasecond diving stimuli compared with general stimuli among the divers is quite reasonable based on the processing principle, which suggests that more efficient information extraction for a given stimulus results in a longer perceived duration.

More interestingly, the RD and RDD results also indicated that the divers reproduced a longer duration for expertise-related stimuli than for general stimuli in subsecond time range, which is believed to be dominated by automatic processing. This result is beyond our initial prediction, since the effect of cognition on processing is smaller at subsecond durations than at suprasecond durations. However, *Rammsayer & Troche (2014)* argued that there is no clear-cut boundary between different mechanisms at different time ranges and it is more appropriate to proceed from the notion of a continuum with regard to the involvement of cognitive processes. Thus, the result noted above can be explained by the fact that sports experts' intense practice improved cognition far beyond what nonexperts can attain. In fact, many studies have already argued that athletes from open-skill sports display

better automatic information processing in coping with high time pressure situations than do nonathletes (*Guldenpenning et al., 2015*; *Guldenpenning et al., 2011*). Similarly, *You et al. (2018)* found that table tennis players outperformed nonathletes in regard to response inhibition even in an unconscious condition where the participants were unaware of the stimulus in the task. Moreover, the findings of *Meng et al. (2019)* also suggested that motor expertise modulates unconscious executive control in table tennis players. These results, combined with the fact that divers, similar to many athletes in open-skill sports, must perform complex movements under high time pressure, suggest that the efficient information processing conferred by their cognitive advantages affects subsecond-duration processing, which is considered to be dominated by automatic or unconscious mechanisms.

In addition, the analysis of the divers' training period and RDD showed that the longer their diving training, the larger the RDD they produced, especially for suprasecond durations. This result is consistent with the suggestions of some researchers that athletes' cognitive advantage in expertise-related tasks is mainly practice dependent and cannot be attributed to innate differences (*Meng et al., 2019*; *Yarrow, Brown & Krakauer, 2009*). In the case of the divers, a longer sport-specific training period resulted in more efficient information processing with regard to expertise-related stimuli, and this more efficient information extraction led these athletes to reproduce longer durations for expertise-related stimuli than for general stimuli. Moreover, the correlation was stronger at suprasecond than subsecond durations, confirming that cognitive processes are more involved in longer durations (*Lewis & Miall, 2003b*; *Rammsayer & Troche, 2014*).

It is important to note that the difference in years of training between the divers and wrestlers ($8.06 \pm 2.84$ versus $2.94 \pm 1.63$) and gender difference between the wrestlers (only 3 females) and the other groups might affect our interpretation of the results. However, the analysis showed no evidence that the group difference in RDD can be attributed to the difference in length of training between divers and wrestlers. For example, the analysis showed consistent results for divers and wrestlers with comparable lengths of training. More importantly, the relationship between training experience and RDD was strengthened when we considered only expertise-related training. Therefore, we supposed that expertise-related training experience played a larger role than general physical training in explaining the increased RDD of divers compared with wrestlers. Additionally, there was little evidence supporting an effect of gender on the RDD results. For example, no significant effect was found in ANOVA with gender as a between-subjects factor. Although there were 2 interactions reached the significance level in our analysis for other dependent variables, the low positive finding rate (2/32) combined with the lack of evidence for the gender effect on duration perception in other studies led us to believe that the current results were solid despite the gender differences between groups.

Regarding the bias results, most participants in the task showed a tendency to overestimate subsecond durations and underestimate suprasecond durations, which is consistent with the results of other study (*Chen, Pizzolato & Cesari, 2014*). This finding is a well-known phenomenon in the field of time perception, especially in the reproduction task, and scholars have already offered a reasonable explanation (*Wearden & Lejeune, 2007*). However, there is also a new perspective based on the Bayesian model, which suggests that

observer integrates a noisy representation of the stimulus with prior information about the stimulus distribution to produce a posterior distribution for the duration to be judged (*Jazayeri & Shadlen, 2010*). More importantly, the group-image type interaction revealed that when the stimuli changed from general to diving stimuli, only the divers overestimated more in both subsecond and suprasecond time ranges. This result combined with the RD and RDD outcomes confirmed that only the divers reproduced longer durations when they viewed expertise-related stimuli compared with general stimuli, while wrestlers and nonathletes with no diving experience showed the opposite result.

The CV and AE ratio results showed that all participants performed better as the stimulus duration grew longer (smaller CV and AE ratio in the suprasecond time range than in the subsecond time range), which is in line with the results of other studies (*Chen & Cesari, 2015*; *Hayashi et al., 2014*; *Lewis & Miall, 2009*). However, previous research found that sports experts have a more precise and stable timing ability than nonathletes, and the results of reduced AE ratio and CV for sports experts were not repeated in this research. For example, *Chen, Pizzolato & Cesari (2014)* found that compared with nonathletes, athletes had reduced AE ratios in subsecond time ranges and reduced CVs in subsecond and suprasecond time ranges for both expertise-related and non-expertise-related stimuli. Our results suggest that there were no differences among the groups with regard to the CV. Regarding the AE ratio, the present study revealed that the divers outperformed the wrestlers and nonathletes in subsecond time range for only general stimuli. Some have argued that the superior precision of duration perception in sports experts can be viewed as a result of their sharpened perceptual-motor system induced by long-term training (*Chen & Cesari, 2015*). Indeed, long-term sports training or motor skill learning experience has been suggested as a factor that induces functional and structural changes in human brain (*Dayan & Cohen, 2011*; *Huang et al., 2018*). Furthermore, many findings have shown that the cerebellum is affected by training experience (*Cannonieri et al., 2007*; *Han et al., 2009*; *Kim et al., 2014*; *Park et al., 2009*). Meanwhile, the cerebellum has been strongly linked to subsecond time perception (*Hayashi et al., 2013*; *Hayashi et al., 2014*; *Lewis & Miall, 2003b*; *Wiener, Turkeltaub & Coslett, 2010*).

Therefore, one possible explanation for the reduced AE ratio found in the diver group is that the divers' cerebellar areas underwent changes induced by intense general physical training, which would improve timing ability in the subsecond time range. Additionally, the wrestlers showed no difference from non-athletes in terms of AE ratios, indicating that their general physical training experience might not be sufficient to upgrade their motor system for improved timing ability in the subsecond time range. Indeed, results from *Sysoeva et al. (2013)* revealed that elite athletes (swimmers and skiers) showed higher accuracy in a tapping task than amateur athletes (wrestlers) or nonathletes. Moreover, *Kim et al. (2014)* found that only elite archers showed increased activity in the cerebellar area in a simulated archery aiming task. However, the divers' advantage in timing precision disappeared in the expertise-related stimulus condition, which is also consistent with our prediction. Since the divers reproduced longer durations for diving stimuli than for general stimuli, the longer durations resulted in larger AE ratios.

There are two possible reasons for the inconsistency between the present results and those of previous studies. First, the task instructions for the participants in previous studies emphasized precision, which might have led different groups (athletes, nonathletes) to take advantage of different strategies (*Chen & Cesari, 2015*). For example, sports experts might be more likely to employ a counting strategy during the task, since the instructions did not forbid counting. This difference could account for the superior performance (i.e., reduced CV and AE ratio) of sports experts (*Rattat & Droit-Volet, 2012*). Second, the sample sizes in previous studies were quite small. For instance, the elite athlete group in *Chen, Pizzolato & Cesari (2014)* had only 12 participants, and there might have been a greater proportion of false positive findings due to the problem of low statistical power (*Fraley & Vazire, 2014*; *Schweizer & Furley, 2016*). However, the lack of additional evidence from other studies examining sports experts' duration perception makes it difficult to draw solid conclusions. Therefore, although the present study showed that sports experts perceive longer durations than nonexperts when viewing expertise-related stimuli, further research is needed to provide more consistent evidence.

Finally, we would like to discuss two potential implications of the findings from this study. First, the perceived duration of expertise-related stimuli can be used to evaluate athletes' information processing efficiency. In the present study, we attributed divers' extended duration perception for expertise-related stimuli to an increased efficiency of information extraction because of their sport-specific (i.e., diving) training experience. Therefore, athletes who perceive longer durations for expertise-related stimuli are likely to have higher information processing efficiency. Information processing efficiency certainly plays a large role in athletic performance, and this inference is supported by the connection between the length of diving training and the extent to which the duration of expertise-related stimuli were overestimated in our research. Thus, the perceived duration of expertise-related stimuli might help coaches select young athletes with great potential. Second, general physical training might be a good supplement to other treatments to correct distortions in time perception and time performance caused by a number of different neurological and psychiatric conditions (e.g., attention-deficit/hyperactivity disorder (ADHD) and autism). Our results showed that sports experts with long-term physical training experience have superior timing ability (i.e., timing precision). This finding is consistent with the work of other researchers and indicates that long-term physical training might improve timing ability. Thus, exercise intervention for people with ADHD or autism might improve their timing ability.

## LIMITATIONS

The current study has three limitations. First, some interpretations of our results must be regarded with caution due to the differences in years of training and gender between the wrestlers and divers. Although our analysis confirmed that the current results and interpretations are reasonable even in light of those factors, experts with compatible lengths of training in different sports will enable researchers to precisely evaluate the effects of expertise-related training and general physical training on duration perception.

Second, the duration reproduction task is not a pure perception task, as the bisection and generalization tasks are. Rather, we made a simplifying assumption that the motor noise (i.e., pressing the button) was constant for all subjects. However, we obtained baseline evaluations of all participants in the general stimulus session and utilized the RDD to evaluate the difference in duration perception for different stimuli in an attempt to ensure that the main results of the present study were not excessively affected by the participants' motor noise. More importantly, the reproduction task was utilized not only because it offered a continuous measurement of duration perception but also because it enabled us to compare our results with those of previous studies. The third limitation is that the correlation analysis between the RDD and the length of diving training in diver group was flawed because it was not a part of our initial data-processing plan. In future studies, a more reasonable sample size for correlation analysis based on a priori power analysis is recommended.

## CONCLUSIONS

The divers reproduced longer durations than wrestlers or nonathletes when they viewed expertise-related stimuli in the temporal reproduction task. This outcome is consistent with our prediction based on the processing principle, which hypothesized that more efficient information extraction would result in longer perceived duration. In addition, the positive correlation between the years of diving training and magnitude of duration overestimation for expertise-related stimuli indicated that sports experts' advantage in information extraction is a result of professional training in the relevant sport. Moreover, this finding also revealed that divers with long-term athletic training experience had more precise duration perception than others in the subsecond time range for general stimuli, suggesting that intense general physical training might improve timing accuracy.

### Funding
The authors received no funding for this work.

### Competing Interests
The authors declare there are no competing interests.

### Author Contributions
- Binbin Jia conceived and designed the experiments, performed the experiments, analyzed the data, prepared figures and/or tables, authored or reviewed drafts of the paper, and approved the final draft.
- Zhongqiu Zhang and Tian Feng conceived and designed the experiments, authored or reviewed drafts of the paper, and approved the final draft.

### Human Ethics
The following information was supplied relating to ethical approvals (i.e., approving body and any reference numbers):

This study received approval from the regional ethics board of the China Institute of Sport Science in China (Approval number: 18-04).

## Data Availability

Data is available in the Supplemental Files and at the Open Science Framework: Jia, Binbin. 2019. "Duration Perception of Chinese Diving Athletes." OSF. October 3. osf.io/83smd.

## Supplemental Information

Supplemental information for this article can be found online at http://dx.doi.org/10.7717/peerj.8707#supplemental-information.

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
