# Peer review of "Sports experts’ unique perception of time duration based on the processing principle of an integrated model of timing"

_PeerJ, doi:10.7717/peerj.8707_

## Round 0.1 · original submission · Major Revisions

I now have received two reviewers' comments. Although both reviewers expressed their interest in your study, several aspects of this manuscript should be revised to improve its clarity. Their observations are presented with clarity so I'll not risk confusing matters by belaboring or reiterating their comments. While I might quibble with the occasional point, I note that I regard the reviewers' opinions as substantive and well-informed. I believe that all of the highlighted reservations, the grouping issues raised by the second reviewer in particular, require contemplation and appropriate attention in revising the document if it is to contribute appropriately to Peerj and the extant literature. Please revise or refute according to the two reviewers' comments and provide a point by point reply in addition to the revised manuscript.

Tsung-Min Hung, PhD., FNAK, FISSP
PeerJ editor
Research chair professor,
Department of Physical Education,
National Taiwan Normal University

Reviewer 1 ·

Basic reporting

No comment.

Experimental design

No comments.

Validity of the findings

No comment.

Additional comments

This study examined whether sports experts (i.e, divers) show longer duration perceptions when viewing stimuli in relation to their expertise domain compared to experts (i.e., wrestler) with other expertise domain and non-experts. Their findings are in line with the study hypothesis by showing that divers reproduced longer durations for diving figures as compared with the neutral figures, while the other groups showed the opposite patterns. More specifically, they also reported a positive correlation between years of training experience and the magnitude of the pronged reproduction duration in the divers.

In my opinion, the manuscript is concisely written, the figures are adequate, and the topic is of interest to PeerJ. However, there are some minor caveats of this study which are needed to be addressed appropriately.

1. It may be beneficial if the author could refer to the recently published work (i.e., Scharfen, & Memmert, 2019) to re-consider the “elite” or “expert” definition for their participants.
2. It is suggested to provide more details about participants’ demographic information that may potentially bias the results of interests such as BMI, physical fitness, the dose of current training.
3. The authors adopted Bayesian statistics for their data without ever making explicit the a priori Bayes Factor they would consider significant or providing a citation that would explain quite what their analysis involved. Please provide more details regarding the use of Bayesian analysis.
4. The discussion reads very well and provides new insights into the expertise-difference in duration perception. I only have one remark: the authors are suggested to propose the potential implications of the current finding to field-based applications.

·

Basic reporting

The paper addressed important topic of time perception and examined rare group of elite athletes that I greatly appreciate. The performed analysis is valid and applied professionally, while the English might be improved at some parts.

My major concerns lays in the group matching. Two important differences between the athlete’s groups were ignored, that needs to be considered. Unfortunately, the athletes’ groups were not matched by gender and, most importantly, years of training. While gender difference between groups is unlikely to affect the interpretation of the results, it needs to be acknowledged and discussed. At the same time, taking into account that wrestler and diving groups are substantially different in the years of training (8.06±2.84 vs 2.94±1.63) the difference in time perception parameters between the groups can be accounted by the general length of training experience, not specific for particular activity (diving or wrestling), as claimed by authors. Indeed, when subjects need to evaluate the duration of picture representing human body (expertise-related stimuli in the current study was a body images) they might involve the body signals that might be important component of time perception (see Wittmann et al., 2013). Years of physical training experience might increase vividness of one’s body representation and lead to longer time perception regardless of the sport you are professional in. In line with this interpretation, the divers time perception results correlated with years of training and those divers who have compatible years of training with wrestlers did not differ from wrestler group (see fig 7). Other evidence to support the “length of training” hypothesis came from AE ratio results from this paper. AE ratio advantage was evident for divers only for general stimuli, but not expertise-related ones, contradicting the initial authors’ hypothesis that a particular expertise leads to changes in time perception, and more consistent with the ideal that, in general, intense physical training leads to changes in time perception irrespective of the field of training (wrestler, diving etc). Such interpretation is also in line with previous research on motor timing (Sysoeva et al., 2013). While smaller AE ratio (time perception precision) and longer duration reproduction for body images in athletes might be unrelated phenomena they might be different sides of the same processess.

Thus, I believe that this study might provide important message to scientific community if takes into account the pointed above limitations, either by better matching the groups or by changing the focus of analysis and interpretation of the results.

Experimental design

The experimental design is valid, the topic is relevant and relatively well defined. Statistical analysis used appropriately, but the problems with group matching by important parameter, such as years of training make the results ambiguous and difficult to interpret.

Validity of the findings

no more to add to the above provided comments

Additional comments

Some recommendation for abstract improvement:
The background section of the abstract need improvement. In the current form it is hard to understand what the authors meant by “integrated timing model” and its prediction.
I’d also like to have the exact duration of stimuli stated in the method section.
As I stated before, the years of training is clearly different between compared athletes group that suggest different interpretation of the results, so this should be also reflected in abstract.

---

## Round 0.2 · accepted · Accept

I have now received all the reviewers’ comment with satisfaction of your reply and revisions from previous comments (I have checked the concerns of R2 and believe your ms is correct in that respect). You and your coauthors have my congratulations. Thank you for choosing PeerJ as a venue for publishing your research work and I look forward to receiving more of your work in the future.

Tsung-Min Hung, PhD., FNAK, FISSP
PeerJ editor
Research chair professor,
Department of Physical Education,
National Taiwan Normal University

Reviewer 1 ·

Basic reporting

No comment

Experimental design

No comment

Validity of the findings

No comment

Additional comments

Based on my previous comments on the manuscript followed by authors' replies and incorporated changes in the latest version of the manuscript, I am fully convinced with these changes/replies. Thanks for the authors' effort. I would recommend publishing this manuscript in its current form.

·

Basic reporting

no comments

Experimental design

I believe there is a mistake in the description of the statistical analysis of additional ANOVA analysis that aimed to examine if the gender differrence between the group influence the observed group diffference in duration reproduction parameters. " Additionally, to evaluate the effect of gender difference on our results, we ran repeated-measures ANOVAs of RD, RDD, bias, CV, and AE ratio results for divers (15 female) and non-athletes (13 female) with both group and gender as the between-subjects factors, time range as the within-subjects factor, and age as the covariate." Author implemented valid approach and decided to examine if the results hold when only females of both groups is considered. However, at the same time they stated that they heve "gender" as between-group factor. How can you have gender as a factor if you have sample of only females? You can examine here the group effect and if it holds here similar as with a full-sample, this is an indicator that gender difference is unlikely to influence the results.

Validity of the findings

no comments

Additional comments

I belive authors undretake signifcant amount of work to examine alternative explanations of their findings that was raised by not optimal group matching. Generally, I satisfied with the current version of the ms, and reccomend it for the publication.